# Leave No TRACE: Black-box Detection of Copyrighted Dataset Usage in Large Language Models via Watermarking

## Abstract

Large Language Models (LLMs) are increasingly fine-tuned on smaller, domain-specific datasets to improve downstream performance. These datasets often contain proprietary or copyrighted material, raising the need for reliable safeguards against unauthorized use. Existing membership inference attacks (MIAs) and dataset-inference methods typically require access to internal signals such as logits, while current black-box approaches often rely on handcrafted prompts or a clean reference dataset for calibration, both of which limit practical applicability. Watermarking is a promising alternative, but prior techniques can degrade text quality or reduce task performance. We propose TRACE, a practical framework for *fully black-box* detection of copyrighted dataset usage in LLM fine-tuning. TRACE rewrites datasets with distortion-free watermarks guided by a private key, ensuring both text quality and downstream utility. At detection time, we exploit the radioactivity effect of fine-tuning on watermarked data and introduce an entropy-gated procedure that selectively scores high-uncertainty tokens, substantially amplifying detection power. Across diverse datasets and model families, TRACE consistently achieves significant detections ($p < 0.05$), often with extremely strong statistical evidence. Furthermore, it supports multi-dataset attribution and remains robust even after continued pretraining on large non-watermarked corpora. These results establish TRACE as a practical route to reliable black-box verification of copyrighted dataset usage.

## 1 Introduction

Large Language Models (LLMs) have demonstrated strong performance across real-world applications, from conversational agents (Thoppilan et al. (2022)) and educational tutoring (Wang et al. (2024)) to medical support (Thirunavukarasu et al. (2023)). Their capabilities stem from pre-training on massive text corpora (Hoffmann et al. (2022)) and, crucially for real deployments, from subsequent *fine-tuning* on smaller, domain-specific datasets curated by enterprises or individual researchers (Wei et al. (2021)). Because such corpora often carry copyright and contractual restrictions, and are typically smaller, carefully curated, and controlled by a single rights holder, potential infringement is especially acute. This underscores the pressing need for verifiable safeguards against unauthorized use (Henderson et al. (2023); Carlini et al. (2021)).

Recent lawsuits against leading AI developers, including OpenAI and Meta, highlight mounting concerns over the unlicensed use of copyrighted material in model training (Reuters (2023b;a); Alter & Harris (2023)). At the same time, commercial models have become increasingly opaque about the composition of their training and adaptation corpora, leaving external observers with little visibility into what data has actually been incorporated (Achiam et al. (2023); Dubey et al. (2024)). Together, these developments underscore the urgency of methods that allow rights holders to independently verify whether their datasets have been used in fine-tuning, thereby enabling dataset copyright protection.

A widely studied approach for detecting training data usage is Membership Inference Attacks (MIAs), which test whether a given sample was part of a model's training set (Shokri et al. (2017)). Existing MIAs can be categorized as white-box, grey-box, or black-box, depending on whether

the detector has full access to model parameters, partial access to intermediate signals such as to-ken probabilities, or only input–output behavior. Likelihood-based methods such as Min-K% (Shi et al. (2023)) exploit token logits under white- or grey-box settings, but such information is rarely available in practice. More recent work on dataset-level inference (Maini et al. (2024)) aggregates multiple MIA scores into a hypothesis test, yet still relies on access to logits and auxiliary non-member data, which limits real-world applicability. Consequently, black-box detection is the most realistic setting. However, current black-box methods typically depend on prompt engineering to elicit verbatim recall of training examples (Karamolegkou et al. (2023); Chang et al. (2023)). For example, DE-COP (Karamolegkou et al. (2023)) reformulates copyright detection as a multiple-choice task over book corpora, but it relies on a clean reference dataset for calibration, which is often impractical.

To determine whether a specific dataset has been used to fine-tune an LLM under a black-box setting, a natural approach is to embed watermarks—imperceptible signals inserted into the training text that later serve as indicators of dataset usage (Lau et al. (2024); Wei et al. (2024); Rastogi et al. (2025)). For example, Wei et al. (2024) inserts random sequences or Unicode lookalikes that are invisible to humans but recognizable by the model. However, existing watermarking techniques often degrade text quality or reduce downstream task performance, which limits their practical applicability to LLMs.

To address these limitations, we propose TRACE (**T**racing watermarked **R**ewriting for **A**ttribution via **C**ut-off **E**ntropy), a novel framework for statistical verification of dataset usage in black-box LLM fine-tuning scenarios. Unlike prior watermarking techniques that compromise utility, TRACE generates *distortion-free* watermarked rewrites of the training corpus using a private key, ensuring that both text quality and downstream task performance are preserved. The watermarked datasets are then released publicly, so that any rights holder can later detect a suspect model. During detection, we build on the insight that fine-tuning on watermarked data induces measurable *radioactivity* in model outputs, and we amplify this effect by selecting high-entropy tokens through an entropy-gated procedure. If we can detect the watermark signal from model outputs using the corresponding key in a statistically significant way, it gives us strong evidence that the model has been fine-tuned on the protected dataset.

Our main contributions are as follows:

- We present TRACE, a novel framework for *fully black-box* detection of dataset usage in LLM fine-tuning. TRACE requires only model input–output interactions, without access to logits or internal parameters.

- We develop an entropy-gated detection procedure that selectively scores high-uncertainty tokens in model outputs, substantially amplifying watermark radioactivity and yielding orders-of-magnitude stronger statistical evidence than existing baselines.

- We show that distortion-free watermarking via rewriting preserves both text quality and downstream utility. Across diverse models and datasets, TRACE consistently achieves highly significant detections ($p < 0.05$), often with overwhelming evidence.

- We further evaluate robustness in extended scenarios. First, we show that TRACE enables *multi-dataset attribution*: when multiple candidate datasets are watermarked with distinct keys, our method reliably attributes fine-tuning to the correct source dataset. Second, we demonstrate robustness to *continued pretraining*: even after further training on large non-watermarked corpora, watermark signals remain detectable with high significance.

## 2 PRELIMINARIES

### 2.1 PROBLEM DEFINITION

We formalize **copyrighted dataset usage detection** as a statistical hypothesis testing problem. Let $M$ be a large language model. Let $\mathcal{Z}$ denote the example space (e.g., input-output pairs) and let $\mathcal{D}$ be the space of a copyrighted dataset. A finite dataset instance is denoted by $D = \{z_i\}_{i=1}^n$, where each example $z_i = (x_i, y_i)$ consists of an input $x_i \in \mathcal{X}$ and output $y_i \in \mathcal{Y}$, with $\mathcal{X}$ and $\mathcal{Y}$ representing the input and output spaces, respectively. The detection task is to determine whether a model $M$ was fine-tuned on $D$. In the black-box setting, we do not observe model parameters or

logits. Instead, we can query $M$ with inputs $\{x_i\}_{i=1}^n$ from $D$ and collect the corresponding outputs $Y^o = \{y_1^o, \ldots, y_m^o\}, y_j^o \in \mathcal{Y}$.

The null hypothesis $H_0$ is that $M$ was *not* fine-tuned on $D$. We define a test statistic that measures the strength of evidence for dataset usage. If the corresponding $p$-value falls below a chosen significance threshold, we reject $H_0$ and conclude that $D$ was likely used; otherwise, we fail to reject $H_0$.

## 2.2 LLM WATERMARKING

Large language models (LLMs) generate text autoregressively, predicting each token based on its preceding context. A watermarking mechanism augments this process by incorporating a watermark key into the sampling procedure. Specifically, the key is combined with the context window through a pseudorandom function to generate a seed that perturbs the token sampling distribution. This seed determines a small logit adjustment (e.g., favoring a subset of tokens) or an equivalent modification of the sampling process, such that the model's outputs remain natural while embedding a hidden statistical signal that can later be verified using the same key.

We employ *SynthID-Text* Dathathri et al. (2024) as the watermarking mechanism in our framework. At each generation step $t$, a pseudorandom seed $r_t$ is derived from the private key $k$ and the context window $c_t = (x_{t-w}, \ldots, x_{t-1})$ of length $w$. This seed drives $d$ independent binary watermark functions, producing *g-values*

$$g_{t,j}(x, r_t) \in \{0, 1\}, \qquad j = 1, \ldots, d, \ x \in V,$$

for each candidate token $x$ in the vocabulary $V$. The model samples candidate tokens from the original distribution and runs a $d$-round tournament: tokens with higher aggregated $g$-values advance through successive rounds, and the final winner becomes the next output token.

For detection, the same key is used to compute watermark statistics on a generated sequence $\mathbf{x} = (x_1, \ldots, x_T)$. Specifically, we measure the empirical win rate of the generated tokens:

$$\text{Score}(\mathbf{x}) = \frac{1}{dT} \sum_{t=1}^{T} \sum_{j=1}^{d} g_{t,j}(x_t, r_t).$$

## 3 METHOD

We propose TRACE, a practical framework that enables dataset owners to verify whether their data has been used to fine-tune large language models (LLMs). Building on the observation of Sander et al. (2024) that models fine-tuned on watermarked text exhibit *radioactivity* in their outputs, TRACE comprises two components (Fig. 1): (i) each dataset owner applies a watermarking function $\mathcal{W} : \mathcal{D} \times \mathcal{K} \to \mathcal{D}$ to release a watermarked rewrite $D' = \mathcal{W}(D, k)$ of their dataset instance $D$ using a private key $k$ (Sec. 3.1); and (ii) the owner employs a black-box detection function $\mathcal{V} : \mathcal{Y} \times \mathcal{K} \to [0, 1]$ on model outputs $\mathbf{y} \in \mathcal{Y}$, yielding a test statistic $\mathcal{V}(\mathbf{y}, k)$ that quantifies the presence of key-aligned watermark signals (Sec. 3.2).

## 3.1 WATERMARKING OF DATASETS

To enable copyrighted dataset usage detection, each dataset owner $i$ generates a watermarked version of their original dataset $D_i \in \mathcal{D}$ using a private watermark key $k_i \in \mathcal{K}$. Concretely, we employ an instruction-tuned LLM to rephrase each sample $(x, y) \in D_i$, where the rephrasing is guided by $k_i$, yielding a rewritten dataset $D_i' = \mathcal{W}(D_i, k_i)$. The resulting $D_i'$ preserves the original task utility while embedding imperceptible statistical watermark signals.

**Watermarking design criteria.** We require four properties for the watermarking process:

- *Semantic fidelity.* Rewriting must preserve the original meaning so that task semantics remain unchanged. With a normalized semantic similarity $S : \mathcal{Y} \times \mathcal{Y} \to [0, 1]$, we require $\mathbb{E}_{(x,y) \sim \mathcal{D}_i} \mathbb{E}_{y' \sim \mathcal{W}((x,y), k_i)} [S(y', y)] \approx 1$.
- *Task preservation.* The rewritten dataset must retain comparable downstream utility. For any training procedure Train and evaluation metric $U$ on test distribution $\mathcal{Q}$, let $M_{D_i} = \text{Train}(D_i)$ and $M_{D_i'} = \text{Train}(D_i')$, we require $U(M_{D_i'}; \mathcal{Q}) \approx U(M_{D_i}; \mathcal{Q})$.

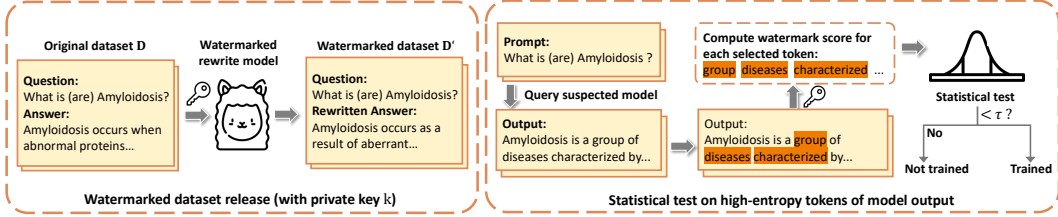

Figure 1: Overview of **TRACE**. The framework has two stages. **(Left)** The dataset owner generates a watermarked rewrite $D' = \mathcal{W}(D, k)$ of the original dataset using a watermarked rewrite model with a private key $k$, and releases $D'$ publicly. **(Right)** To verify dataset usage, the owner queries a suspect model $M$ with prompts and collects outputs. High-entropy tokens are selected against the private key to compute the watermark scores. A statistical test is then conducted to decide whether the model exhibits watermark radioactivity, indicating that it was fine-tuned on $D'$.

- *Key-specific radioactivity.* The detection signal must be specific to the owner's key: only if the suspect model has been trained on $D'_i = \mathcal{W}(D_i, k_i)$ should the statistical test return a significant result.

- *Distributional neutrality (distortion-free).* For any context $h$, averaging over random keys recovers the original next-token distribution: $\mathbb{E}_{K \sim \mathcal{K}}\big[\mathcal{W}(\cdot \mid h, K)\big] = \mathcal{W}(\cdot \mid h, \varnothing)$. This neutrality ensures that text quality remains unchanged, since no key-agnostic bias toward particular tokens or patterns is introduced.

To instantiate $\mathcal{W}$ in our framework, we adopt *SynthID-Text* ( Dathathri et al. (2024)), a distortion-free watermarking algorithm developed by Google DeepMind and deployed in production systems. SynthID-Text embeds statistical traces directly into the sampling process while preserving the base model's distribution. We empirically verify in Sec. 4.2 that SynthID-Text meets the above design criteria.

## 3.2 DETECTION OF DATASET RADIOACTIVITY

Given a black-box suspect model $M$ and the owner's private key $k$, we test whether $M$ was fine-tuned on the owner's watermarked dataset $D' = \mathcal{W}(D, k)$. The datasets we consider vary in format. For question–answer datasets, which are the primary setting in our experiments, we directly use the original question as the prompt. When answers are too short (e.g., multiple-choice datasets with single-token answers) or when the dataset is not in question–answer form (e.g., long-form copyright text), we instead construct prompts from the watermarked portions of $D$ and cast the task as continuation writing. Prompt templates are provided in Appendix G. This allows us to elicit training-like generations $Y = \{y_1, \ldots, y_m\}$ from $M$.

After obtaining the outputs $Y$, we compute watermark scores for each generated token. Using the private key $k$, every token position $t = 1, \ldots, T$ in a sequence $\mathbf{y} = (x_1, \ldots, x_T)$ is mapped to a depth vector $\mathbf{g}_t = \big(g_{t,1}(x_t, r_t), \ldots, g_{t,d}(x_t, r_t)\big)$, where $g_{t,i} \in \{0, 1\}$. For stronger detection ability, we collapse $\mathbf{g}_t$ to a scalar score via a depth-weighted average (Dathathri et al. (2024)):

$$\bar{g}_t \;=\; \frac{1}{d} \sum_{i=1}^{d} w_i \, g_{t,i}(x_t, r_t), \qquad w_i \propto d + 1 - i, \;\; \sum_{i=1}^{d} w_i = d.$$

**Motivation.** During fine-tuning, a model tends to update more at positions where its next-word prediction is uncertain—i.e., several continuations are similarly plausible. Watermarking methods that slightly nudge sampling are most effective in such uncertain positions: when the base distribution is sharp (one token clearly dominates), the watermark rarely changes the outcome; when it is dispersed, small key-guided nudges more often change which token is chosen Kirchenbauer et al. (2023). Consequently, the model learns key-specific preferences primarily at uncertain positions, and the watermark signal ("radioactivity") concentrates there after fine-tuning. At detection time, including many highly confident tokens mostly adds noise, so we focus our detector on the most uncertain tokens.

We use token-level entropy of an auxiliary model as a simple, robust proxy for uncertainty. For each generated position t with history $h_t$, let $\hat{q}_t(\cdot)$ be the next-token distribution and define the entropy

$$H_t = -\sum_{w \in \mathcal{V}} \hat{q}_t(w) \log \hat{q}_t(w).$$

We pool all output tokens across prompts, rank them by $H_t$ in descending order, and apply a hard entropy gate: keep the top q% highest-entropy tokens. The selected tokens form the scored set $\mathcal{S}$, on which we compute watermark scores g-values and perform a statistical test. Algorithm 1 summarizes the full procedure.

To determine whether the suspect model exhibits watermark radioactivity, we construct a one-sided hypothesis test. For each selected token $t$, the average watermark score $\bar{g}_t$ has expected value $1/2$ under the null hypothesis $H_0$ (the model was not fine-tuned on the owner's dataset). This allows us to form a standardized test statistic

$$Z = \left(\frac{1}{|\mathcal{S}|}\sum_{t \in \mathcal{S}} \bar{g}_t - \frac{1}{2}\right)\sqrt{4\, d_{\text{eff}}\, |\mathcal{S}|},$$

where $d_{\text{eff}} \triangleq d^2 / \sum_{i=1}^{d} w_i^2$ accounts for weighting across watermark rounds $d$. The statistical significance of detection is then quantified by the one-sided p-value $p = 1 - \Phi(Z)$, with $\Phi(\cdot)$ the standard normal CDF. A small p-value indicates that the outputs contain watermark signal aligned with the private key, consistent with the model having been fine-tuned on the watermarked dataset.

## 4 EXPERIMENTS

### 4.1 EXPERIMENTAL SETUP

We evaluate the effectiveness of TRACE by conducting the experiments across different models and datasets in section 4.2, and perform ablation studies in section 4.3.

**Dataset and Models.** We evaluate on diverse datasets covering the main formats of fine-tuning corpora: question–answer pairs with natural language responses (GSM8k (Cobbe et al. (2021)), Med (Ben Abacha & Demner-Fushman (2019)), Alpaca (Taori et al. (2023)), Dolly (Conover et al. (2023))), multiple-choice datasets with single-token answers (MMLU (Hendrycks et al. (2009)), ARC-C (Clark et al. (2018))), and pure text corpora without QA structure (Arxiv (face (2022))). The dataset statistics are shown in Appendix B. We use Llama-3.2-3B-Instruct Meta (2024a), Phi-3-mini-128k-instruct Microsoft (2024), and Qwen2.5-7B-Instruct Qwen (2024) as target models, and generate watermarked datasets with Llama-3.1-8B-Instruct Meta (2024b) using prompts, which are shown in Appendix G.

**Implementation details.** For watermark text generation, we set the sampling parameters to temperature $= 0.8$, top-$p = 0.95$, and top-$k = 50$. We use temperature $= 0.5$ and top-$p = 0.9$ to obtain the model output in the detection stage. For QA datasets, we perform supervised fine-tuning with LoRA, a learning rate of $1.0 \times 10^{-4}$ for Llama-3B and Phi-3.8B (or $5.0 \times 10^{-5}$ for Qwen-7B), and 3 training epochs. For text corpora, we adopt continued pre-training with a learning rate of $5.0 \times 10^{-5}$ for 2 epochs. The details are shown in Appendix C.

**Evaluation Metrics.** We use the p-value as the primary metric to quantify the statistical significance of dataset usage detection. To evaluate the quality of rewritten (watermarked) text, we report P-SP (Wieting et al. (2021)) and perplexity (PPL). For downstream task performance, we employ different metrics based on the answer format: accuracy for datasets with reference answers that allow exact matching, and BERTScore (Zhang et al. (2019)) together with ROUGE (Lin (2004)) for open-ended generation tasks, where no single ground-truth answer exists. The details of metrics are provided in Appendix D.

**Baselines.** We compare our method against two categories of baselines: (i) the black-box detection method DE-COP (Duarte et al. (2024)), and (ii) Grey-box detection methods: three sample-level MIAs: PPL, Min-K% Prob (Shi et al. (2023)), Zlib (Carlini et al. (2021)), and DDI (Maini et al. (2024)). To adapt the sample-level MIAs to our dataset-level setting, we treat the training set as positive samples and the evaluation set as negative samples. For DE-COP, we measure the detection accuracy on training versus evaluation data. A detailed description of baseline implementations and p-value computation is provided in Appendix E.

Table 1: P-values for copyrighted dataset–usage detection across methods. Lower values indicate stronger evidence; $p < 0.05$ denotes significance (bold). TRACE consistently yields smaller $p$-values than baselines on all four datasets. $p$-values of TRACE are computed on the 40k highest-entropy positions selected from 100k generated tokens (i.e., fewer than 1000 complete responses when grouped as samples); baselines use 1000 responses for comparability.

| Category | Method | Med | | | Dolly | | |
|---|---|---|---|---|---|---|---|
| | | Llama-3B | Phi-3.8B | Qwen7B | Llama-3B | Phi-3.8B | Qwen7B |
| Grey-box | PPL | 0.03 | 0.23 | **0.04** | 0.69 | 0.62 | 0.28 |
| | Zlib | 0.07 | 0.16 | 0.06 | 0.20 | 0.38 | 0.31 |
| | Min-K | **0.02** | 0.08 | **0.02** | 0.37 | 0.55 | 0.80 |
| | DDI | 0.12 | 0.19 | **0.01** | 0.29 | **0.02** | **7.6e-05** |
| Black-box | DE-COP | **1.8e-03** | 0.85 | 0.30 | 0.82 | 0.47 | **7.53e-04** |
| | TRACE | **2.0e-172** | **0.05** | **6.4e-83** | **7.5e-32** | **0.03** | **8.8e-09** |
| Category | Method | Alpaca | | | GSM8k | | |
| | | Llama-3B | Phi-3.8B | Qwen7B | Llama-3B | Phi-3.8B | Qwen7B |
| Grey-box | PPL | 0.51 | 0.79 | 0.88 | 0.30 | **2.7e-05** | **3.9e-22** |
| | Zlib | 0.89 | 0.82 | 0.74 | 0.80 | 0.11 | **5.3e-08** |
| | Min-K | 0.27 | 0.61 | 0.44 | 0.10 | **8.7e-11** | **7.2e-52** |
| | DDI | 0.19 | 0.01 | 0.81 | **1.5e-02** | **2.0e-10** | **9.9e-44** |
| Black-box | DE-COP | 0.09 | 0.36 | **9.5e-06** | 0.15 | 0.20 | 0.06 |
| | TRACE | **2.2e-94** | **1.0e-11** | **1.4e-49** | **1.5e-44** | **3.1e-40** | **7.0e-36** |

Table 2: P-values for continuation-style detection across three datasets. For each setting, we generate 100k tokens and apply entropy gating to select a fixed budget of 10k high-entropy tokens for scoring.

| Dataset | MMLU | ARC-C | Arxiv |
|---|---|---|---|
| P-value | 2.1e-07 | 1.5e-07 | 7.6e-06 |

## 4.2 MAIN RESULTS

**Detection Results.** Table 1 reports $p$-values for copyrighted dataset–usage detection across four datasets and multiple baselines. TRACE consistently attains vanishingly small $p$-values, providing overwhelming evidence that the suspect models were fine-tuned on the watermarked data. Grey-box MIAs (PPL, Zlib, Min-K%, DDI) exhibit mixed performance: while some achieve significance on GSM8k and occasionally on Med or Dolly for specific models, they fail to generalize across model families. The black-box baseline DE-COP reaches significance in a few cases but remains inconsistent. In comparison, TRACE yields dramatic improvements over DE-COP, with geometric mean gains of approximately $8.95 \times 10^{83}$ on Med, $2.45 \times 10^{12}$ on Dolly, $1.00 \times 10^{49}$ on Alpaca, and $3.81 \times 10^{38}$ on GSM8k. Overall, TRACE is uniformly significant across all datasets and models, underscoring its robustness and reliability.

We also evaluate continuation-style detection on text-only and single-token multiple-choice corpora (Table 2). TRACE remains statistically significant across all four datasets, with particularly strong significance on three of them. Although the signal is somewhat weaker than on the QA datasets, it still provides robust evidence of dataset usage. The representative examples of watermarked text and model output detected are shown in Appendix H.

**Text Quality Preservation.** We evaluate whether watermarking affects text quality using two metrics: Perplexity (PPL), which measures fluency, and semantic similarity (P-SP, Wieting et al. (2021)), which captures semantic preservation. As shown in Table 3, rewritten datasets not only match the perplexity of the originals but in several cases achieve even lower PPL. They also attain high semantic similarity (P-SP between 0.85–0.91), indicating that watermarking has minimal impact on text quality.

Table 3: Quality preservation of rewritten text across datasets, evaluated by Perplexity (PPL) and semantic similarity (P-SP). Lower PPL indicates better fluency, while higher P-SP reflects stronger semantic consistency with the original text.

| **Metric** | | Med | GSM8k | Dolly | Alpaca |
|---|---|---|---|---|---|
| Perplexity | Original | 9.50 | 14.75 | 16.88 | 8.54 |
| | Rewritten | 9.14 | 9.47 | 11.51 | 8.50 |
| P-SP | – | 0.91 | 0.87 | 0.85 | 0.90 |

Table 4: Downstream performance on evaluation dataset. Generation tasks are evaluated with BERTScore and ROUGE-1 (↑ higher is better), while GSM8K is additionally assessed with Accuracy (↑).

| Model | Method | Generation Quality | | | | | | QA Accuracy | | |
|---|---|---|---|---|---|---|---|---|---|---|
| | | Med | | Dolly | | Alpaca | | GSM8k | | |
| | | BS | RG | BS | RG | BS | RG | BS | RG | Acc |
| Llama-3B | Original | 0.62 | 0.32 | 0.61 | 0.32 | 0.70 | 0.48 | 0.62 | 0.34 | 0.61 |
| | Unicode | 0.37 | 0.10 | 0.35 | 0.07 | 0.39 | 0.14 | 0.54 | 0.18 | 0.54 |
| | TRACE | 0.62 | 0.30 | 0.61 | 0.33 | 0.68 | 0.45 | 0.58 | 0.33 | 0.61 |
| Phi-3.8B | Original | 0.62 | 0.30 | 0.65 | 0.41 | 0.73 | 0.53 | 0.80 | 0.61 | 0.73 |
| | Unicode | 0.43 | 0.13 | 0.36 | 0.06 | 0.41 | 0.14 | 0.57 | 0.21 | 0.63 |
| | TRACE | 0.50 | 0.17 | 0.64 | 0.39 | 0.71 | 0.49 | 0.68 | 0.51 | 0.72 |
| Qwen-7B | Original | 0.66 | 0.37 | 0.67 | 0.45 | 0.50 | 0.25 | 0.77 | 0.59 | 0.75 |
| | Unicode | 0.38 | 0.11 | 0.36 | 0.09 | 0.41 | 0.16 | 0.63 | 0.34 | 0.61 |
| | TRACE | 0.64 | 0.35 | 0.64 | 0.40 | 0.71 | 0.50 | 0.65 | 0.45 | 0.80 |

**Downstream Performance.** Table 4 shows that across all three model families, the Unicode (Wei et al. (2024)) baseline substantially degrades generation quality and often reduces QA accuracy. In contrast, TRACE closely tracks the Original models, which are trained on the original dataset, typically within ±0.04 on BERTScore/ROUGE-1, while preserving or improving QA accuracy on GSM8k.

**False Positive Analysis.** Under the null: models not fine-tuned on the watermarked data, and tested with each dataset's own key to the model outputs for the entropy-gated detection process, all 12 model–dataset pairs yield $p$-values above 0.05 (minimum = 0.10), indicating no statistically significant detections and demonstrating TRACE's robustness to false positives (Table 5).

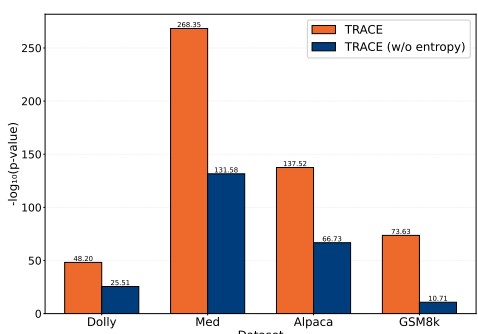

Figure 2: Entropy ablation experiments across four datasets using 40k scored tokens with the top 70% by entropy on LLaMA-3B.

### 4.3 FACTORS INFLUENCING DETECTION POWER

To better understand the conditions under which TRACE is more effective, we conduct ablation studies on factors that influence detection strength.

**Entropy Gating.** Fig. 2 compares TRACE with and without entropy gating. Entropy gating boosts detection on every dataset. Without entropy gating, the test is still significant in most cases (well above the dotted line), but the evidence is far weaker—especially on GSM8K and Dolly. These results demonstrate the effectiveness of entropy-gated detection: by concentrating the test on high-entropy positions, TRACE amplifies key-aligned watermark signal.

Table 5: False-positive analysis under the null (no watermarked fine-tuning). Entries are $p$-values.

| Dataset | Llama-3B | Phi-3.8B | Qwen-7B |
|---------|----------|----------|---------|
| Med     | 0.46     | 1.00     | 1.00    |
| Dolly   | 1.00     | 0.98     | 1.00    |
| Alpaca  | 0.34     | 0.76     | 0.77    |
| GSM8k   | 0.55     | 1.00     | 0.10    |

Table 6: Multi-dataset attribution results: $\log_{10}(p\text{-value})$ of detection using different keys on a model fine-tuned with 2 watermarked datasets. Stronger evidence (more negative) is expected along the diagonal.

|              | Med          | GSM8k        |
|--------------|--------------|--------------|
| $k_{Med}$    | **3.2e-129** | 1.0          |
| $k_{GSM8k}$  | 1.0          | **5.8e-45**  |

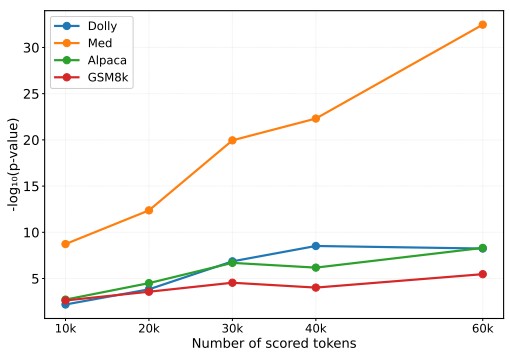

(a) $-\log_{10}(p)$ vs. number of scored tokens with $\rho = 50\%$ watermarked samples.

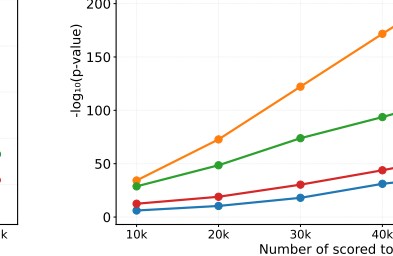

(b) $-\log_{10}(p)$ vs. number of scored tokens with $\rho = 100\%$ watermarked samples.

Figure 3: Effect of watermarked-sample proportion and number of scored tokens on detection strength for TRACE. Curves plot $-\log_{10}(p)$ (higher = more significant) as a function of the number of scored tokens. (a) The proportion of watermarked samples: $\rho = 50\%$. (b) $\rho = 100\%$.

**Proportion of watermarked training data.** Fig. 3a vs. Fig. 3b illustrates how detection strength scales with the fraction of watermarked samples used in fine-tuning at every token budget. We observe that TRACE already exhibits strong radioactivity when only 50% of the training set is rewritten, with $-\log_{10}(p)$ surpassing 30 on Med at 60k token budgets. This implies that practitioners need not rewrite the entire dataset: watermarking just proportion of the training data is sufficient to achieve highly significant detection, while leaving the remaining unmodified to minimize disruption to the original dataset.

**Token budget.** We further analyze how the number of tokens used in detection impacts statistical power. As shown in Fig. 3a and Fig. 3b, detection strength increases steadily as more tokens are allocated. Even with 10k tokens, all datasets already exceed the significance threshold. As the budget grows, detection becomes rapidly stronger.

## 5 EXTENDED ANALYSES

### 5.1 MULTI-DATASET ATTRIBUTION

We conduct this experiment to test whether TRACE can not only detect dataset usage but also attribute it to the correct source when multiple candidate datasets are involved. By assigning distinct watermark keys $\{k_j\}_{j=1}^{J}$ to each dataset, we expect that a model fine-tuned on a particular dataset will exhibit radioactivity only under its corresponding key, while other keys should yield no signal. Table 6 confirms this intuition: for the model jointly fine-tuned on Med ($k_{Med}$) and GSM8k ($k_{GSM8k}$), taht is, $J = 2$, the strongest evidence appears along the diagonal entries, with highly significant $p$-values of $3.2 \times 10^{-129}$ and $5.8 \times 10^{-45}$ respectively. In contrast, off-diagonal entries remain near 1.0, indicating no detectable watermark signal when the wrong key is used. This diagonal–off-diagonal separation demonstrates that TRACE is able to reliably attribute fine-tuning to the correct dataset among multiple candidates.

## 5.2 CONTINUED PRETRAINING ROBUSTNESS

We evaluate whether continued pretraining on additional non-watermarked corpora diminishes radioactivity detection. Starting from the model fine-tuned on the watermarked Alpaca dataset, we further pretrain it on OpenOrca (Lian et al. (2023)), a large corpus containing 2.94 million samples. TRACE remains highly significant after this continued pretraining: the $p$-value shifts from $2.2 \times 10^{-94}$ to $2.6 \times 10^{-36}$. Although the signal is weaker than before, it remains far below the $p=0.05$ threshold, demonstrating that TRACE is robust to continued pretraining.

## 6 RELATED WORK

**Membership Inference Attack.** Membership inference attacks ask whether a single sample was in a model's training set, originating with shadow-model attacks Shokri et al. (2017). For LLMs, popular instantiations include loss/perplexity thresholding and difficulty calibration Yeom et al. (2018); Watson et al. (2021), reference-based LiRA variants Mireshghallah et al. (2022a;b), and token-level refinements such as MIN-K% Shi et al. (2023). These methods often assume gray-box access (logprobs), suitable reference models, or distribution-matched validation data, which are not practical at LLM scale. While black-box evaluation is the most realistic setting, existing approaches such as DE-COP still depend on a clean reference dataset to calibrate their multiple-choice preference tests, limiting their applicability when such baselines are unavailable Karamolegkou et al. (2023).

**Dataset inference.** Recent work has shifted from single-sample membership to document- or dataset-level attribution. One line of work embeds artificial patterns into text—such as fictitious trap sequences (Meeus et al. (2024b)) or imperceptible Unicode substitutions (Wei et al. (2024)) to test whether models exhibit a preference for them, though such modifications risk reducing readability and decreasing model performance. Meeus et al. (2024a) uses token-level logit statistics aggregated across an entire document, which requires gray-box access and is less practical. Dataset Inference method Maini et al. (2024) combines multiple weak MIAs into a statistical test, yet it depends on IID reference sets and log-prob access, making it sensitive to distribution drift. More recently, Waterfall embeds watermark patterns via paraphrasing for robust attribution, though its effectiveness diminishes on shorter texts (Lau et al. (2024)).

**LLM Watermarking and Radioactivity.** Watermarking schemes for LLMs modify token probabilities or sampling procedures to embed statistical traces in generated text Kirchenbauer et al. (2023); Dathathri et al. (2024); Chen et al. (2025) in either distortion-free or distortion-based ways. These techniques were originally designed for output attribution, ensuring that AI-generated text can be distinguished from human-written content. Sander et al. (2024) showed that watermark signals indeed contaminate fine-tuned LLMs. While radioactivity arises as an unintended side effect of watermarking, it provides a promising direction for dataset-level usage detection in realistic black-box settings.

## 7 CONCLUSION

We introduced TRACE, a practical framework for *fully black-box* verification of copyrighted dataset usage in LLM fine-tuning. TRACE watermarks datasets via distortion-free rewriting and then performs entropy-gated statistical testing on model outputs, requiring only input–output access. Across four datasets and three model families, TRACE delivers uniformly significant detections and achieves orders-of-magnitude stronger evidence than existing baselines, while preserving text quality and downstream task performance. Ablations show that entropy gating is crucial for amplifying signal and that watermarking only ∼50% of the training data already yields pronounced radioactivity, offering a favorable utility–enforcement trade-off. Extended analyses demonstrate reliable multi-dataset attribution using distinct keys and robustness to continued pretraining on large, non-watermarked corpora.

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

## A  ALGORITHMS

---

**Algorithm 1:** Entropy-Gated Radioactivity Test

---

**Input:** Model $M$; key $k$; prompts $\mathcal{P}$; selection rule: top-$q\%$
**Output:** $p$-value

**Query.** Collect outputs $\mathcal{Y} = \{y_t\}_{t=1}^{N_0}$ by querying $M$ with $\mathcal{P}$.;
**for** $t = 1$ **to** $N_0$ **do**
  Compute entropy $H_t$ for $y_t$;
  Using $k$, compute watermark score $\mathbf{g}_t = (g_{t,1}, \ldots, g_{t,d})$ and their weighted average $\bar{g}_t$;

**Rank.** Form the list $\mathcal{L} = \{(t, H_t, \bar{g}_t)\}$ and sort by $H_t$ in descending order.;

**Select.** Set $B = \lfloor q/100 \cdot N_0 \rfloor$ and take the top-$B$ items of $\mathcal{L}$ as the scored set $\mathcal{S}$.;

**Test.** Construct a test statistic based on $\mathcal{S}$ and convert it into $p$-value.

---

## B  DATASET STATISTICS

| Dataset | #Samples | Total Tokens | Watermark Field | Field Tokens | Field Ratio |
|---|---|---|---|---|---|
| Med | 14,766 | 4,229,172 | Answer | 4,019,980 | 95.05% |
| Dolly | 13,509 | 2,347,866 | response | 1,056,576 | 45.00% |
| Alpaca | 46,584 | 7,288,892 | output | 6,437,305 | 88.32% |
| GSM8k | 7,473 | 1,177,960 | answer | 749,674 | 63.64% |

We calculate all token counts using the tokenizer of Llama-3.1-8B-Instruct for consistency across datasets and model families. The Watermark Field column denotes which part of each dataset (e.g., output, answer) is rewritten with the watermarking model. Only these fields are watermarked during dataset construction.

## C  IMPLEMENTATION DETAILS

All experiments are conducted on NVIDIA L40 GPUs. For SynthID-Text watermarking, we use $n\text{-}gram = 2$, $m = 30$ tournament layers, and a *Bernoulli*$(0.5)$ $g$-value distribution, which follows the configuration of Dathathri et al. (2024). Watermarked text is generated with sampling parameters temperature $= 0.8$, top-$p = 0.95$, and top-$k = 50$. During the detection stage, we adopt a more conservative decoding setup with temperature $= 0.5$ and top-$p = 0.9$.

For all datasets used in our experiments, we employ standard supervised fine-tuning (SFT) with LoRA. Llama-3.2-3B-Instruct and Phi-3-Mini-128k-Instruct are fine-tuned with a learning rate of $1 \times 10^{-4}$, whereas Qwen-2.5-7B-Instruct, being larger, uses a learning rate of $5 \times 10^{-5}$. All three model families are trained for 3 epochs under the same set of hyperparameters, including a warmup ratio of 0.03, a per-device batch size of 4, gradient accumulation of 8 steps, and a cosine learning-rate scheduler.

To study robustness under continued pretraining, we take the model already fine-tuned on the watermarked Alpaca dataset and further pretrain it on the OpenOrca corpus (2.94M samples). This additional stage involves roughly 6000 training steps, compared with the 2184 steps used for Alpaca fine-tuning.

## D  METRIC DETAILS

**P-SP.** The Paraphrase Semantic Proximity (P-SP) metric Wieting et al. (2021) is a semantic similarity model trained on large-scale paraphrase data. It is designed to distinguish true paraphrases from unrelated text. Formally, given original text $x$ and rewritten text $y$, P-SP computes the cosine similarity between their embeddings:

$$\text{P-SP}(x, y) = \cos\bigl(g(x),\, g(y)\bigr),$$

where $g(\cdot)$ is the P-SP encoder. We use P-SP to measure semantic preservation by comparing each watermarked answer with its original counterpart (higher is better).

**Perplexity (PPL).**  Perplexity evaluates how well a language model predicts a text sequence, with lower values indicating higher model confidence. For a sequence $S = (s_1, \ldots, s_N)$ under model $\theta$, perplexity is defined as

$$\mathrm{PPL}_\theta(S) = \exp\!\left(-\tfrac{1}{N}\sum_{i=1}^{N}\log P_\theta(s_i \mid s_{<i})\right).$$

We compute PPL of rewritten answers with respect to the base model to quantify fluency and detect possible degradation introduced by watermarking.

**BERTScore.**  BERTScore Zhang et al. (2019) compares contextual embeddings from a pretrained transformer model to evaluate the similarity between a candidate and a reference. It captures both lexical and semantic alignment beyond surface token overlap. In our experiments, we compute BERTScore between rewritten answers and original answers to evaluate the downstream model performance (higher is better).

**ROUGE.**  ROUGE Lin (2004) measures overlap of $n$-grams or longest common subsequences between candidate and reference text. We report ROUGE-L to evaluate lexical consistency between rewritten answers and their original versions, complementing embedding-based metrics (higher is better).

# E  BASELINE DETAILS

We briefly describe the baselines used in our experiments.

1. *Perplexity (PPL).* A classic loss-based MIA that scores each text by the average token negative log-likelihood under the suspect model; lower values indicate higher familiarity. We use PPL to separate the *training* split (positives) from the *evaluation* split (negatives) and assess significance at the split level.

2. *Min-K% Prob.* A reference-free detector that, for each sequence, selects the $k\%$ *lowest-probability* (i.e., *highest-loss*) tokens and averages their negative log-likelihoods; members tend to contain fewer extremely low-probability tokens. We aggregate per-example scores over the training/evaluation splits for dataset-level inference. We set $k\% = 0.2$ in the experiment.

3. *Zlib.* Following prior practice, we use the sequence's zlib compression length as a complexity proxy to calibrate likelihood (intuitively discounting highly compressible strings), and then compare the calibrated scores between splits.

4. *DDI.* Our implementation of dataset inference that linearly combines multiple reference–free, loss–based features and performs a split–level statistical test. For each sequence we compute PPL, mean_logp, kmin_logp (lowest $10\%$ token log–probs), kmax_logp (highest $10\%$), zlib_ratio (compressed/Raw length), and length. We apply per–feature winsorization (2.5% tails) and standardization, then randomly split Suspect/Validation into A/B halves: on A we learn a linear combiner (least–squares regression) that maps features to a membership score (Suspect=0, Val=1); on B we score all samples and run a one–sided Welch t–test ($H_1 : \mu_{\text{Suspect}} < \mu_{\text{Val}}$).

5. *DE-COP.* A black-box probe that constructs four-way MCQs mixing the verbatim original answer with three paraphrases; models trained on the target text are more likely to choose the verbatim option. For our dataset-level setting, we build MCQs for each split and compare verbatim-option accuracies between training and evaluation. Note that we include DE-COP as a representative baseline for passive copyright detection. DE-COP performs detection on the original, unmodified data, whereas TRACE acts as a proactive protection mechanism that modifies the dataset prior to release.

We repurpose sample-level MIAs to a dataset-level decision by treating the *training* dataset as positives and the *evaluation* dataset as negatives. The null hypothesis states that the model shows no difference in familiarity between the two datasets. We therefore use a one-sided Welch's $t$-test (robust to unequal variances and sample sizes) to compare means:

$$t = \frac{\bar{x}_{\text{train}} - \bar{x}_{\text{eval}}}{\sqrt{\frac{s^2_{\text{train}}}{n_{\text{train}}} + \frac{s^2_{\text{eval}}}{n_{\text{eval}}}}},$$

where $\bar{x}$, $s^2$, and $n$ denote the sample mean, variance, and size. For loss-like metrics (smaller $\Rightarrow$ more member-like), we test the one-sided alternative $\bar{x}_{\text{train}} < \bar{x}_{\text{eval}}$.

For the black-box DE-COP baseline, each example yields a four-way MCQ consisting of the verbatim original answer and three paraphrases (rewritten by the same model used in our pipeline). We measure the model's accuracy on the training and evaluation datasets and compare the proportions via a two-proportion $z$-test with pooled variance:

$$z = \frac{p_{\text{eval}} - p_{\text{train}}}{\sqrt{\hat{p}(1 - \hat{p})\left(\frac{1}{n_{\text{eval}}} + \frac{1}{n_{\text{train}}}\right)}}, \qquad \hat{p} = \frac{x_{\text{eval}} + x_{\text{train}}}{n_{\text{eval}} + n_{\text{train}}},$$

where $p_{\text{train}}$ and $p_{\text{eval}}$ are observed accuracies, $n_{\text{train}}$ and $n_{\text{eval}}$ are sample sizes, and we use the one-sided alternative $H_1 : p_{\text{eval}} < p_{\text{train}}$ (a model trained on the dataset should recognize verbatim originals more reliably on the training split). To mitigate option-position bias, we enumerate all $4! = 24$ permutations of answer ordering per item and report permutation-level proportions as our primary result.

## F  PROMPT TEMPLATE FOR REWRITING DATASET WITH WATERMARK

---

**Prompt Template for Rephrasing Med**

Rephrase the following medical answer of the question: **{Question}**. Preserve all medical facts and information. Do NOT add personal opinions or extra sentences. Provide ONLY the rephrased answer.

Original Answer: **{Answer}**

Rephrased Answer:

---

**Prompt Template for Rephrasing Dolly**

Instruction: **{Instruction}**
Context: **{Context}**

Rephrase the following response of the question. Original Response: **{Response}**

Task: Please rephrase the above response, preserving all facts and key information. Do NOT add personal opinions or extra content. Provide ONLY the rephrased response.

Rephrased Response:

---

**Prompt Template for Rephrasing Alpaca**

Instruction: **{Instruction}**
Input: **{Input}**

Original Response: **{Response}**

Task: Please rephrase the above response, preserving all facts and key information. Do NOT add personal opinions or extra content. Provide ONLY the rephrased response.

Rephrased Response:

---

**Prompt Template for Rephrasing GSM8k**

Rephrase the following solution explanation of the math question: **{Question}**.
Preserve all numerical and logical details. Do NOT add personal opinions or extra sentences. Provide ONLY the rephrased explanation.

Original Explanation:**{CoT}**

Rephrased Explanation:

---

**Prompt Template for Rephrasing ARC-C**

Rephrase the following multiple-choice question: **{Question}**

Preserve all factual information. Do NOT add personal opinions or extra sentences. Choices must remain exactly the same text and labels, DO NOT rewrite them.

**{Choices}**

Rephrased Question:

---

**Prompt Template for Rephrasing MMLU**

Rephrase the following multiple-choice question while preserving all factual information and key details.
Do NOT add personal opinions or extra sentences. Provide ONLY the rephrased question (without rewriting the choices).

Original Question: **{Question}**

Choices: **{Choices}**

Rephrased Question:

---

## G  PROMPT TEMPLATE FOR CONTINUING THE DATASET OUTPUT

**Prompt Template for Rephrasing ARC-C and MMLU**

You are a continuation restorer. The following is the beginning of a question that appeared in your training data. Your task is to recall and reconstruct the missing portion. Continue from the cutoff point; do NOT repeat the beginning and do NOT provide options or answers.

Truncated question: **{Content}**

Continue:

---

**Prompt Template for Rephrasing Arxiv**

You are a continuation restorer. The following is the beginning of an abstract that appeared in your training data. Your task is to recall and reconstruct the missing portion. Continue from the cutoff point; do NOT repeat the beginning and do NOT provide options or answers.

Truncated abstract: **{Content}**

> Continue:

# H  REPRESENTATIVE EXAMPLES

We present representative examples from all datasets used in our experiments, comparing the original text, the watermarked text, and the model-generated text after fine-tuning. All model outputs shown in this section are generated by the watermark-text–finetuned version of Llama-3.2-3B-Instruct.

## H.1  GSM8K

**Question:**
Ann's favorite store was having a summer clearance. For $75 she bought 5 pairs of shorts for $7 each and 2 pairs of shoes for $10 each. She also bought 4 tops, all at the same price. How much did each top cost?

**Original Answer:**
Ann spent $75 at the store. The cost of 5 pairs of shorts at $7 each is calculated as 5 x $7 = $35. The cost of 2 pairs of shoes at $10 each is found by 2 x $10 = $20. The combined cost of the shorts and shoes is $35 + $20 = $55. Given that Ann spent $75 in total, subtracting the cost of the shorts and shoes leaves a remaining cost of $75 - $55 = $20. This remaining cost is the total amount she spent on 4 tops, so to find the cost of each top, we divide $20 by 4, resulting in $5. #### 5

**Watermarked Answer:**
Ann spent $75 at the store. The cost of 5 pairs of shorts at $7 each is calculated as 5 x $7 = $35. The cost of 2 pairs of shoes at $10 each is found by 2 x $10 = $20. The combined cost of the shorts and shoes is $35 + $20 = $55. Given that Ann spent $75 in total, subtracting the cost of the shorts and shoes leaves a remaining cost of $75 - $55 = $20. This remaining cost is the total amount she spent on 4 tops, so to find the cost of each top, we divide $20 by 4, resulting in $5. #### 5

**Model Output:**
To determine the cost of each top, we first calculate the total amount Ann spent on shorts and shoes. Five pairs of shorts at $7 each totals $35, and two pairs of shoes at $10 each totals $20. Combining these amounts gives us a total of $55 spent on shorts and shoes. Since Ann spent a total of $75, we subtract the amount spent on shorts and shoes from the total to find the amount spent on tops: $75 - $55 = $20. Given that Ann purchased 4 tops, we divide the amount spent on tops by the number of tops to find the cost per top: $20 / 4 = $5.

## H.2  MED

**Question:**
Is complete LCAT deficiency inherited?

**Original Answer:**
This condition is inherited in an autosomal recessive pattern, resulting in both copies of the gene in each cell having mutations. Typically, parents of an individual with this condition are carriers of one copy of the mutated gene each and usually do not display symptoms.

**Watermarked Answer:**
This condition is inherited in an autosomal recessive pattern, resulting in both copies of the gene in each cell having mutations. Typically, parents of an individual with this condition are carriers of one copy of the mutated gene each and usually do not display symptoms.

**Model Output:**
Complete LCAT deficiency is inherited in an autosomal recessive pattern, resulting in both copies of the gene in each cell bearing mutations. Typically, parents of an individual with complete LCAT deficiency are carriers of one copy of the mutated gene each and usually do not display symptoms.

## H.3 ALPACA

**Instruction:**
Detect whether the following statement is a metaphor, a simile, or neither.

**Input:**
She was like a ray of sunshine.

**Original Answer:**
The statement uses a comparison with a descriptive phrase starting with "like," making it a simile.

**Watermarked Answer:**
The statement uses a comparison with a descriptive phrase starting with "like," making it a simile.

**Model Output:**
This statement is a simile. A simile is a figure of speech that compares two unlike things using the words "like" or "as." In this case, the statement compares the person to a ray of sunshine, highlighting their bright and cheerful nature.

## H.4 DOLLY

**Instruction:**
How long do bears hibernate?

**Content:**
Bears use shelters, such as caves and logs, as their dens; most species occupy their dens during the winter for a long period of hibernation, up to 100 days.

**Original Response:**
Bears can hibernate up to 100 days in their dens during the winter.

**Watermarked Response:**
Bears typically occupy their dens, such as caves and logs, for a prolonged period of up to 100 days during winter.

**Model Output:**
Bears hibernate for up to 100 days. They use caves and logs as their dens, often occupying them for extended periods during the winter. This prolonged hibernation is a natural adaptation to conserve energy and survive the harsh winter conditions.

## H.5 ARC-C

**Question:**
A ship leaks a large amount of oil near a coastal area. Which statement describes how the oil most likely will affect the coastal habitat?

**Choices:**
A. Fish reproduction rates will increase.
B. Water birds will be unable to use their wings.
C. Water plants will be exposed to more sunlight.
D. Coastal plants will have access to more nutrients.

**Watermarked Question:**
A ship with a significant oil leak occurs in a coastal area. What effect on the coastal habitat is most likely to occur due to the oil spill?

**Model Output:**
would the oil leak have on the marine ecosystem in the area, and what steps would be taken to mitigate the damage.

## H.6 MMLU

**Original Question:**
Lawyers Abel and Baker are the members of the law partnership of Abel and Baker in a small town that has only one other lawyer in it. Abel and Baker do a substantial amount of personal injury work. Client was severely and permanently injured in an automobile collision when struck by an automobile driven by Motorist. Client employed the Abel and Baker firm to represent her in obtaining damages for her injuries. At the time Client employed Abel and Baker, the statute of limitations had six weeks to run on her claim. The complaint was prepared but not filed. Abel and Baker each thought that the other would file the complaint. The statute of limitations ran on Client's claim against Motorist. Client has filed suit against Abel and Baker for negligence. That case is on trial with a jury in a court of general jurisdiction. "In addition to proving that Abel and Baker were negligent, Client must establish, as a minimum, that she"

**Choices:**
A. would have, but for her lawyers' negligence, recovered from Motorist.
B. had a good faith claim against Motorist that was lost by her lawyers' negligence.
C. was severely and permanently injured when struck by Motorist's automobile.
D. did not negligently contribute to the failure to have the complaint filed.

**Watermarked Question:**
To prevail in her negligence claim against Abel and Baker, the Client must demonstrate that their negligence in handling her case caused her to lose her legitimate opportunity to recover damages from Motorist.

**Model Output:**
that Abel and Baker owed a duty of care to the Client.

## H.7 ARXIV

**Title:**
Improved conformalized quantile regression

**Original Abstract:**
Conformalized quantile regression is a procedure that inherits the advantages of conformal prediction and quantile regression. That is, we use quantile regression to estimate the true conditional quantile and then apply a conformal step on a calibration set to ensure marginal coverage. In this way, we get adaptive prediction intervals that account for heteroscedasticity. However, the aforementioned conformal step lacks adaptiveness as described in (Romano et al., 2019). To overcome this limitation, instead of applying a single conformal step after estimating conditional quantiles with quantile regression, we

propose to cluster the explanatory variables weighted by their permutation importance with an optimized k-means and apply k conformal steps. To show that this improved version outperforms the classic version of conformalized quantile regression and is more adaptive to heteroscedasticity, we extensively compare the prediction intervals of both in open datasets.

**Watermarked Abstract:**
Conformalized quantile regression combines the benefits of conformal prediction and quantile regression, using quantile regression to estimate the true conditional quantile and then incorporating a conformal step on a calibration set to ensure marginal coverage, resulting in adaptive prediction intervals that take into account heteroscedasticity. However, applying a single conformal step to estimated conditional quantiles from quantile regression, as previously described, lacks the adaptiveness addressed in (Romano et al., 2019). To address this limitation, we introduce an alternative approach, dividing explanatory variables by their permutation importance, applying an optimized k-means clustering, and using k conformal steps in succession. This enhanced version is compared extensively in open datasets against the classic conformalized quantile regression, demonstrating its superiority and greater adaptability in handling heteroscedasticity through improved prediction intervals.

**Model Output:**
lack of adaptiveness in the prediction intervals, as it relies on a fixed conformal radius that does not account for the varying levels of heteroscedasticity across different quantiles. To address this limitation, we propose a new approach that incorporates a conformal step on a calibration set that is specifically designed to adapt to the heteroscedasticity of the quantile regression estimates. This calibration set is constructed by selecting a subset of data points that are representative of the quantile regression estimates and then applying a conformal step to this subset. The resulting prediction intervals are then used to guide the construction of the final prediction intervals. Our approach allows for a more adaptive and accurate prediction of the conditional quantiles, while also ensuring marginal coverage.

