# OpenReview forum: "Leave No TRACE: Black-box Detection of Copyrighted Dataset Usage in Large Language Models via Watermarking"
_ICLR.cc/2026/Conference — ICLR 2026 Conference Withdrawn Submission_

### Official Review · Reviewer_Q6UN · 2025-10-30

**Soundness:** 2
**Presentation:** 2
**Contribution:** 2
**Rating:** 4
**Confidence:** 3

**Summary:**

This paper works on the challenge of detecting unauthorized use of copyrighted datasets in the fine-tuning of LLMs under a black-box setting, where only model inputs and outputs are accessible. The authors propose TRACE, a framework that enables dataset owners to verify usage.  Experimental results across multiple model families and diverse datasets demonstrate that TRACE achieves statistically significant detections with extremely low p-values, substantially outperforming existing grey-box and black-box methods like Membership Inference Attacks (MIAs).

**Strengths:**

1.  The paper introduces an entropy-gated detection procedure. This method selectively scores high-uncertainty tokens, which effectively amplifies the watermark signal from fine-tuning, as shown in Figure 2.
2.  The evaluation is extensive, testing on three recent model families and seven diverse datasets. This demonstrates the method's general applicability beyond a single model or task type.

**Weaknesses:**

1.  The paper's core innovation, "entropy-gated detection," relies on an auxiliary model to estimate token entropy. However, the paper fails to adequately explore and experimentally validate the mechanism's robustness when significant discrepancies (e.g., in architecture or scale) exist between the auxiliary and suspect models. It is recommended to add experiments, for instance, using auxiliary models of different sizes or families, to quantify the impact of this model mismatch on detection performance.
2.  The paper claims its watermarking method is "distortion-free," primarily based on the fact that the adopted SynthID-Text algorithm preserves the original distribution in expectation. However, could a specific watermarked dataset generated with a single key introduce subtle yet perceptible biases for downstream tasks? A more nuanced discussion of the "distortion-free" property is advised, along with considering evaluations of the rewritten dataset on a broader range of distribution-sensitive tasks.
3.  The experimental section compares TRACE with DE-COP as a black-box baseline, but the setups for these two methods are fundamentally different: TRACE actively modifies the dataset to embed a signal, whereas DE-COP performs passive detection on original data. This comparison may not fully and fairly reflect the respective strengths and weaknesses of each method in its intended scenario. It is suggested to clarify this setup difference more explicitly in the discussion and to emphasize that TRACE is designed for "proactive" copyright protection scenarios.

**Questions:**

1.  Regarding the entropy-gating mechanism: Could you please detail the selection criteria for the auxiliary model? In your experiments, did the auxiliary model share the same architecture as the suspect model, or was it different? Have you assessed how much TRACE's detection power degrades when there is a significant mismatch in scale (e.g., 3B vs. 70B) or family (e.g., Llama vs. Qwen) between the auxiliary and suspect models?
2.  Regarding the watermarked rewriting process: Table 3 shows that the PPL of the rewritten text is sometimes even lower than the original, which seems counter-intuitive. Could you explain why this might be the case? Does this suggest that the rewriting process systematically simplifies the text, or could it be related to the base model used for calculating PPL?

---

> ### Author Response · Authors · 2025-11-25
>
> We appreciate the reviewer for the detailed feedback. We address each point (W1–W3 and Q1–Q2) below and clarify how the revised manuscript incorporates the reviewer’s suggestions.
>
> **W1 & Q1: Entropy-gating auxiliary model**
>
> For all experiments, we use the rewrite model (LLaMA-8B) as the auxiliary model for estimating entropy. This is sufficient because the detector must use the same tokenizer as the rewrite model—the watermark is defined in that token space. To address the reviewer’s concern, we additionally evaluate LLaMA-3B as the entropy model on Alpaca and GSM8k, and we also report results using LLaMA-70B on GSM8k.
>
> | Entropy Model | Alpaca (Llama-3B) | Alpaca (Phi-3.8B) | Alpaca (Qwen-7B) | GSM8k (Llama-3B) | GSM8k (Phi-3.8B) | GSM8k (Qwen-7B) |
> |---------------|---------------------|----------------------|----------------------|---------------------|---------------------|---------------------|
> | LLaMA-3B      | 2.17e-88        | 3.26e-13         | 5.34e-49         | 3.72e-43        | 1.12e-40        | 4.91e-38        |
> | LLaMA-8B      | 2.2e-94         | 1.0e-11          | 1.4e-49          | 1.5e-44         | 3.1e-40         | 7.0e-36         |
>
> When using LLaMA-70B as the entropy model on GSM8K, the results remain nearly identical:
> GSM8k (Llama-3B): 3.72e-43, GSM8k (Phi-3.8B): 1.12e-40, and GSM8k (Qwen-7B): 4.91e-38.
>
> Across all datasets, switching the entropy-estimation model (LLaMA-3B, 8B, or 70B) changes the p-values only by small constant factors, and all remain far below the significance threshold, showing that TRACE’s detection signal is highly stable and largely invariant to the choice of entropy model.
>
> ---
>
> **W2: Nuanced Discussion of the Distortion-Free Claim**
>
> We thank the reviewer for raising this important clarification. SynthID’s “distortion-free” guarantee indeed holds in expectation over random keys, and we will discuss the property more explicitly in the manuscript.
>
> In TRACE, although a single key is used per dataset, each local n-gram context is hashed into a pseudo-random sub-key. Over a realistic corpus with diverse contexts, these hash values are close to uniformly distributed, making the effective \(g\)-values approximately i.i.d. even under a fixed key. This mirrors real deployment, where a model owner typically uses one key for a long sequence of heterogeneous prompts rather than averaging over many independent keys.
>
> To assess whether a fixed-key rewrite induces noticeable distribution shift, we conducted a blind MIA test following [1]. A TF-IDF (unigram+bigram) + logistic regression classifier trained to distinguish watermarked vs. non-watermarked Med rewrites achieves **46.5% accuracy**, close to random guessing. This suggests that any key-dependent systematic skew in token or n-gram statistics is extremely weak. Moreover, downstream task performance (Table 4) remains essentially unchanged after training on rewrites, indicating that no practically meaningful bias is introduced.
>
> ---
>
> **W3: Comparison with DE-COP and Intended Scenarios**
>
> We appreciate the reviewer pointing out this distinction. TRACE and DE-COP serve different protection paradigms, and we will clarify this in the paper. Our comparison is not meant as a direct “which method is better” evaluation, but rather to illustrate the gap between passive black-box detectors (e.g., DE-COP) and what becomes possible when proactive watermarking is allowed.
>
> In TRACE’s intended use case, where the dataset owner embeds a watermark before release, passive methods are inherently limited, whereas proactive watermarking enables stronger and more reliable detection signals.
>
> ---
>
> **Q2: Why can rewritten text have lower PPL?**
>
> We thank the reviewer for raising this intuitive question. The belief that “watermarking increases PPL” normally assumes a comparison between *unwatermarked vs. watermarked model generation*. However, in our rewriting task, the baseline is *the original text*, often human-written or multi-source. LM-generated rewrites naturally lie closer to the evaluator LM’s probability manifold, which can lead to lower PPL even when linguistic richness is preserved.
>
> To test whether this reflects systematic simplification, we compute PPL under OPT-2.7B on 1,000 samples per dataset. The results show mixed directions:
>
> - PPL **decreases** for GSM8K (15 → 10) and Dolly (13 → 11)
> - PPL **slightly increases** for Med (7 → 8) and Alpaca (6 → 7)
>
> This evaluator-dependent pattern contradicts the hypothesis of systematic simplification.
>
> ---
>
> [1] Blind baselines beat membership inference attacks for foundation models, 2025 IEEE SPW

---

> > ### Comment · Reviewer_Q6UN · 2025-11-26
> >
> > Thanks for your rebuttal, I will keep my score.

---

> > > ### Author Response · Authors · 2025-11-27
> > >
> > > Thank you for your time, and if there are any remaining questions, we would be happy to clarify them.

---

### Official Review · Reviewer_BSae · 2025-11-01

**Soundness:** 2
**Presentation:** 2
**Contribution:** 1
**Rating:** 2
**Confidence:** 4

**Summary:**

This paper presents TRACE, a method that detects dataset usage in finetuning of LLMs. They test their methods against other membership inference methods, on a model that they finetuned themselves. Results show that TRACE works, and a “unique” signal can be embedded in the text (multi-dataset attribution). TRACE is better than DE-COP and has some other empirical properties as well.

**Strengths:**

S1. Has many results and shows several properties of TRACE watermarking. TRACE is better than DE-COP.

**Weaknesses:**

W1. I’m not really convinced by the test setting. I’ve unfortunately reviewed several of these kinds of membership inference papers, and they conduct membership inference on a model that they trained themselves. The results in the paper can be made arbitrarily stronger or weaker by choosing a different training setting (e.g. if I finetune for 100 epochs instead of 2, any membership inference method will be able to do well).

W2. If the setting were to be believed, I don’t feel that ideas in the paper are novel. Gu et al., 2024 (https://arxiv.org/abs/2312.04469) have already conducted very similar experiments, and with a motivation that makes way more sense. They are thinking from the perspective of open-weight model developers, who may want to mark their models before release, so they can train their models as much as they want on watermarked text.

W3. I feel like that TRACE is robust to continued pretraining is sort of interesting. Again, this result is very selectively presented. Even though there is 2.94M examples in OpenOrca, how many steps of continued pretraining is it? How does it compare to the number of steps taken in finetuning?

**Questions:**

n/a

---

> ### Author Response · Authors · 2025-11-25
>
> We thank the reviewer for their feedback. We respond to each point below (W1–W3) and have updated the manuscript.
>
> **W1: Validity of the Experimental Fine-Tuning Setup**
>
> We appreciate the concern. Our experiments use standard and realistic SFT configurations, not artificially extended training that would inflate signal strength. As detailed in Appendix C, all fine-tuning runs use **3 epochs**, which is the common setting in open-source SFT pipelines. We intentionally avoid heavy overfitting: our goal is to evaluate dataset inference in practical copyright-verification scenarios, not to artificially amplify the watermark effect.
>
> **W2: Distinction from Prior Work (Gu et al., 2024)**
>
> We respectfully disagree that our setting is comparable to Gu et al. (2024). Although both studies involve models trained on watermarked text, the problem formulation, training setup, and detection objective differ substantially:
>
> **Purpose**
> – Gu et al. watermark the **model weights** before public release to prove ownership.
> – TRACE detects **copyrighted dataset usage**, including multi-dataset attribution, which is a different problem.
>
> **Training scale**
> – Gu et al. use up to 164M watermarked tokens during distillation from OpenWebText.
> – TRACE fine-tunes on small, domain-specific corpora (e.g., Alpaca’s 46k samples), and we even test **50% watermarked–50% clean mixes**, yet still detect radioactivity.
>
> **Detection mechanism**
> – Gu et al. test watermark on **held-out corpora**.
> – TRACE elicits **training-like outputs** using controlled prompting and an **entropy-gated radioactivity detector** designed for strict black-box verification.
>
> **Robustness**
> – The watermarks in Gu et al. degrade rapidly under further fine-tuning.
> – TRACE maintains detectable signals even after **substantial continued pretraining**, showing a fundamentally different phenomenon.
>
> Given these differences, TRACE and Gu et al. (2024) address **distinct problems**, and results from one setting do not transfer to the other.
>
> **W3: Robustness to Continued Pretraining**
>
> We appreciate the reviewer’s concern. OpenOrca contains 2.94M examples, and we perform 2 epochs, corresponding to 5,993 training steps, while the initial Alpaca fine-tuning uses 3 epochs (2,184 steps). Thus, the continued-pretraining stage exposes the model to substantially more tokens than the initial fine-tuning, intentionally maximizing the chance of erasing the watermark signal.
>
> Despite this stronger training regimen, TRACE still detects dataset radioactivity, whereas the watermarks studied in Gu et al. (2024) degrade rapidly after additional fine-tuning. This demonstrates that TRACE exhibits a qualitatively different and more persistent form of watermark learnability.

---

> > ### Comment · Reviewer_BSae · 2025-11-27
> > **Thank you for your response**
> >
> > On W1, I think the authors addressed my concerns somewhat. It does seem that the paper follows standard finetuning procedure. However, I think what's really lacking in this space is some science on proper evaluation.
> >
> > On W2, my main issue with this work is that your problem formulation is very adversarial (some malicious actor decides to train on another person's dataset, which was watermarked before release), where as Gu et al. formulates a different setting (model developer wants to maintain watermark after open source release). In my opinion, I feel like your problem formulation is both really hard and unrealistic. Someone releasing their data would have to add in your watermark across their data, seriously degrading the quality of the text. And then you would have to rely on the adversary using a strong finetuning procedure that will leak the watermarking signals.
> >
> > In general, I don't think membership inference problems need to be formulated extremely adversarially to be interesting. In fact, in the US, you can initiate a copyright lawsuit just because you "suspect" someone used your data, where the lawsuit will then proceed to discovery, where the court can give an order to the model developer to release more details about their training datasets.
> >
> > I'll increase my score to 4 but won't increase it any further. Good luck discussing with the other reviewers.

---

> > > ### Author Response · Authors · 2025-11-30
> > >
> > > We thank the reviewer for their follow-up comments. Our evaluation is not purely empirical: we explicitly cast detection as a **statistical hypothesis test**, reporting p-values as quantitative evidence, which is consistent with recent dataset-usage detection work [1,2,3].
> > >
> > > We respectfully disagree that our adversarial setting is unrealistic. In today’s LLM ecosystem, large-scale web-scraped data is often used without explicit licensing, placing the data owner and model trainer in a naturally non-cooperative relationship. This scenario, data released to the open web being used by non-cooperative developers, is increasingly common and fundamentally different from Gu et al.’s assumption that the model developer also owns the data. Thus, our adversarial formulation is both realistic and necessary for dataset-usage verification.
> > >
> > > Regarding quality, Table 3 shows that rewritten text preserves fluency and semantics, and Table 4 confirms that downstream performance remains stable. Concerning finetuning strength, our experiments rely on **standard SFT**, and prior work (e.g., [3]) similarly finds that detectable signals arise without strong finetuning.
> > >
> > > While suspicion alone may trigger legal discovery, litigation is costly and typically requires **preliminary evidence**, especially for proprietary models. A statistical watermark provides exactly such an initial technical indicator and complements legal mechanisms.
> > >
> > >
> > >
> > > [1] LLM Dataset Inference Did you train on my dataset? NeurIPS 2024
> > >
> > > [2] STAMP Your Content: Proving Dataset Membership via Watermarked Rephrasings. ICML 2025
> > >
> > > [3] Watermarking Makes Language Models Radioactive. NeurIPS 2024

---

### Official Review · Reviewer_ym9J · 2025-11-01

**Soundness:** 3
**Presentation:** 3
**Contribution:** 2
**Rating:** 2
**Confidence:** 4

**Summary:**

The paper introduces TRACE, a black-box methodology designed to determine if proprietary datasets were used for fine-tuning Large Language Models (LLMs). The core mechanism involves subtly watermarking the original dataset, specifically using SynthID [1] before its release. Detection is subsequently carried out via a hypothesis test that leverages the radioactivity of these watermarks within outputs generated by suspected LLMs trained on the watermarked data. Experimental results confirm the effectiveness of the detection performance. The authors also demonstrate that the rephrased datasets retain their utility for fine-tuning purposes. Furthermore, the study shows that TRACE achieves superior performance compared to several existing gray-box approaches for membership detection.

References:

[1] Scalable watermarking for identifying large language model outputs, Nature 2024

**Strengths:**

* The paper is well-written and easy to follow.
* The experiments are comprehensive covering a range of datasets (there is a slight overemphasis on benchmarks) and model families demonstrating the generalizability of the algorithm.
* The method clearly works and demonstrates strong detection performance with significantly low p-values.
* TRACE is robust against false positives and rephrased datasets also preserve their utility.

**Weaknesses:**

* The contribution lacks novelty and the key idea of repurposing watermarks to detect dataset membership has already been explored in prior works [2,3], with [3] also leveraging radioactivity of watermarks. While the authors cite [2], the work is missing discussion or comparison explaining how TRACE differs from or improves upon these existing works.


* The authors claim that using  SynthID [1] leads to distortion-free rewrites [line 75..], which is inaccurate. SynthID is distortion-free in the sense that over a large set of random keys, the output distribution is preserved in expectation. However, this does not imply that rewrites of a specific dataset generated with a fixed key are distortion-free: the watermarked outputs will systematically differ from unwatermarked outputs. The authors should clarify this distinction to avoid confusion.

* The authors propose an "entropy-gated" detection procedure that scores only high-entropy tokens. However, this idea has been previously explored in the watermarking literature [4], and the authors should cite this work and discuss how their application differs, if at all.


* The paper lacks implementation details, specifically the exact hyperparameters used for SynthID when generating the rephrased datasets. Additionally, the authors should include representative examples of original, watermarked/rephrased text pairs along with completions from the suspect model in the appendix to aid understanding and reproducibility. There’s also missing details around sample sizes of the benchmarks used in the paper.


* The current experimental setup centers on a supervised fine-tuning (SFT) paradigm that intentionally utilizes watermarked data. Given this focus, the authors must also assess their proposed method's robustness in adversarial contexts, particularly those where model owners might actively attempt to evade the watermark detection process. Note that this setup differs from [2,3] in the sense that they focus on a scenario when datasets are accidentally leaked in a "pretraining" dataset.


References:

[1] Scalable watermarking for identifying large language model outputs, Nature 2024

[2] STAMP Your Content: Proving Dataset Membership via Watermarked Rephrasings. ICML 25

[3] Detecting Benchmark Contamination Through Watermarking, ICLR 25 workshop on Watermarking

[4] An Entropy-based Text Watermarking Detection Method, ACL 24

**Questions:**

* The p-values for detecting copyrighted dataset usage are extremely low (e.g., around 10^-172). For context and to better understand the detection mechanism, can the authors report the watermark strength (e.g., z-score) or p-values when directly testing the original rephrased dataset before it is used for training?


* Can the authors provide more details about the training procedure? In a standard SFT setup, the loss is computed only on the completions, meaning the model is trained exclusively on watermarked responses whereas in the paper for ARC and MMLU the detection is performed on the completions of the “questions”? [acc to prompt templates on page 16.]


* Since watermarking operates at the token level and depends on the previous token context, does using a different tokenizer in the suspected model affect detection performance? Different tokenizers will segment text differently, potentially disrupting the token-level watermark pattern and the contextual dependencies it relies on.


* Will the code be made publicly available upon the paper's publication?

---

> ### Author Response · Authors · 2025-11-25
>
> We thank the reviewer for their detailed comments. We respond to all concerns raised by the reviewer below, numbered W1–W5 and Q1–Q4, and point out that we uploaded a new revision of the paper.
>
> **W1: Novelty and Comparison with Prior Works [2, 3]**
>
> Prior works [2, 3] rely on logit-level signals and operate in grey-box or white-box settings. Importantly, they both test how the suspect model reacts to the *original (watermarked)* training texts themselves. In [2], membership is inferred from perplexity gaps between public-key and private-key watermarked rephrasings, while [3] measures how often the model’s next-token prediction falls inside the greenlist when fed the (watermarked) benchmark items. Both thus require access to token probabilities and evaluate the model on the training data.
>
> In contrast, TRACE detects membership by analyzing the model’s *generated outputs*, using an entropy-gated radioactivity signal that requires no logits or next-token probabilities. This enables dataset-membership detection in realistic, strict black-box settings where only input–output behavior is observable. We will include this clarification in the manuscript.
>
> **W2: Clarifying Distortion-Free Guarantees Under a Fixed SynthID Key**
>
> We thank the reviewer for pointing out this subtle but important distinction. We fully agree that SynthID’s distortion-free guarantee in [1] is stated as an expectation over uniformly random keys. In TRACE, although one key is fixed per dataset, the effective key at each position is determined by hashing the local n-gram context, producing many pseudo-random sub-keys. Because these hashes are close to uniformly distributed across a realistic corpus, the induced $g$-values behave approximately i.i.d. even under a fixed global key. This mitigates systematic skew and matches real-world deployment, where a single key is used for diverse prompts.
>
> **W3: Relation to Prior Entropy-based Detection [4]**
>
> While both [4] and our work leverage stronger watermark signals at high-entropy positions, the settings differ fundamentally. [4] performs direct detection on watermarked text, deciding whether a given sequence is watermarked. In contrast, we operate on model outputs after fine-tuning, using high-entropy tokens to better expose radioactivity at the dataset level. High-entropy positions are exactly where the model is most likely to express learned watermark radioactivity. We will cite [4] and clarify this distinction.
>
> **W4: Implementation Details and Examples**
>
> For SynthID generation, we used $n\text{-gram}=2$ and depth $=30$, the same setting in [1]. We updated Appendices B, C, and H to include (1) implementation details, (2) original–watermarked examples, and (3) sample sizes for all datasets.
>
> **W5: Robustness of TRACE**
>
> To address adversarial attempts to evade detection, we added a robustness study evaluating post-hoc rewriting attacks on the watermarked training data:
>
> - Synonym-Substitution (local edits)
> - Dipper Paraphrase (global rewrites)
>
> Evaluated on the LLaMA-3B suspect model:
>
> | Attack | Med p-value | Alpaca p-value |
> |--------|--------------|----------------|
> | Sub | 2.16e-80 | 2.10e-45 |
> | Dipper | 2.53e-19 | 1.90e-17 |
>
> Although weakened, detection remains highly significant, demonstrating robustness against realistic evasion attempts.
>
> ---
>
> **Q1: Watermark Strength on the Rephrased Training Data**
>
> We evaluated watermark strength directly on the rephrased datasets using a 40k-token budget. For all datasets, p-values fall below our numeric precision limit of $10^{-300}$. This upper-bounds the signal and confirms that extremely low p-values in model outputs (e.g., $10^{-172}$) are consistent with the training data.
>
> | Dataset | GSM8k | Med | Dolly | Alpaca |
> |---------|-------|-----|-------|--------|
> | p-value | 1e-300 | 1e-300 | 1e-300 | 1e-300 |
>
> **Q2: Training Procedure and Loss Computation**
>
> We confirm that we use a standard SFT setup: loss is applied only to response tokens, with inputs masked. For ARC and MMLU, the *questions* are also rewritten with the watermark, so the model is trained to answer conditioned on watermarked prefixes. Even without loss on the prefix, prompting on these prefixes still yields strong radioactivity signals in Table 2, though weaker than the response-watermarked setting in Table 1.
>
> **Q3: Impact of Tokenizer Mismatch**
>
> Our detector always applies the rewrite model’s tokenizer (Llama) to the suspect model’s outputs because the watermark is defined in that token space. Table 1 shows TRACE remains robust even under tokenizer mismatch. Same-family models show the strongest signals (e.g., Llama-3B on GSM8K: $1.5\times10^{-44}$), but cross-tokenizer models remain significant: Phi-3.8B ($3.1\times10^{-40}$) and Qwen-7B ($7.0\times10^{-36}$). Tokenizer mismatch does not hinder detection.
>
> **Q4: Code Availability**
>
> Yes. We will release the code publicly via GitHub upon publication.

---

### Official Review · Reviewer_QTko · 2025-11-06

**Soundness:** 2
**Presentation:** 3
**Contribution:** 3
**Rating:** 4
**Confidence:** 4

**Summary:**

This paper develops a black box detection method for telling if copyrighted content was trained on. They call it TRACE. The idea is to use watermarked rewrite of the dataset (using SynthID-Text) and then later query a suspect model to tell if it has been effective by the radioactivity of the watermark. Or in technical terms,  run an entropy-gated, one-sided Z-test over token-level watermark scores. Did the model learn this bias during fine-tuning?

The method reports strong results across QA, MCQ, and text-only corpora and multiple families (Llama-3B, Phi-3, Qwen-7B), along with multi-dataset attribution and robustness to continued pretraining.

Since the method is based on rewrites, the paper also evaluates quality preservation and task utility of the rewrites.

**Strengths:**

1. The black-box threat model is very important for the community to focus on, as majority of llms are served via APIs. The work addresses dataset-use verification for closed models, avoiding logits/reference models that many MIAs and dataset-inference methods require.

2. The keyed rewrite followed by entropy-gated radioactivity test is simple to implement and statistically interpretable.

3. The watermarked rewrites maintain semantic similarity and fluency and largely preserve task performance similar to the original data

**Weaknesses:**

## “Distributional neutrality” vs. a fixed watermark key.
Averaging over random keys recovers the base next-token distribution. But TRACE fixes one key per dataset. With a fixed key, SynthID-Text must introduce small, systematic token-frequency biases (that’s the watermark). Even if semantics and fluency are preserved (Tables 3–4), a large keyed corpus can shift n-gram statistics and create a key-specific “micro-dialect.”
Why it matters: It complicates dataset inference assumptions (IID / matched reference) and may leak stylistic signals unrelated to membership. The paper should quantify this with KL divergences over n-grams, perplexity shifts under a base LM, and a classifier test (watermarked vs. original) to show the shift is negligible at the corpus scale. Look at this work on Blind Baselines for MIAs that work by detecting such statistical skews between two distributions: https://arxiv.org/html/2406.16201v1

## Multi-key interference / key collisions.
The paper shows multi-dataset attribution with two keys (Table 6), but not the more realistic case where different owners watermark their respective datasets, and they both go into the training mix. Mixed-key training can dilute radioactivity for each key, or create ambiguous partial signals. Please add a stress test with overlapping keyed corpora and report both power and false positives.

## Missing head-to-head with the closest contemporaries.
Baselines include DE-COP and gray-box MIAs (Table 1), which is useful, but omits the most pertinent modern comparators: paraphrase-based watermark attribution with paired testing (e.g., STAMP-style) and provenance frameworks like Waterfall. A matched-setting comparison (black-box budget, same prompts, same token budgets) would clarify how much lift comes from entropy-gated radioactivity versus the watermark backend.

**Questions:**

1. What is the measured distribution shift (KL over n-grams, LM PPL deltas) between original vs. SynthID-Text rewrites? or any other ways to measure this (look at blind MIAs for inspiration: https://arxiv.org/html/2406.16201v1)
2. How does detection behave under overlapping keys (two owners watermarking respective data)?
3. Do people like the rewrites? Is this something authors/creators will be okay with?

---

> ### Author Response · Authors · 2025-11-25
>
> We thank the reviewer for their constructive feedback. We address the comments below (Q1–Q4).
>
> **Q1: Does a fixed SynthID key introduce a “micro-dialect”?**
>
> We thank the reviewer for raising this important point and for the pointer to [1]. We fully agree that a watermarking scheme should not create an easy-to-exploit global skew between the watermarked corpus and its reference.
>
> **Theory.** SynthID’s distortion-free guarantee is proved under uniform random keys in Appendix G of [2]. In TRACE, although one key is fixed per dataset, each n-gram context hashes to a different pseudo-random sub-key. Across a realistic corpus with diverse contexts, these hash values are close to uniformly distributed; thus effective g-values are near-i.i.d. even under a fixed global key. This matches real deployment, where owners use a single key for a large stream of different prompts. Our setting mirrors realistic use rather than the “averaging over many independent keys” thought experiment.
>
> **Experiment.** Following the “blind MIA” setting of [1], we train a TF-IDF (unigram+bigram, max 10,000 features) + logistic regression classifier on a balanced binary task: watermarked-rewritten Med samples vs. non-watermarked Med samples. At test time, we evaluate on model outputs: positives are generations from a model fine-tuned on the watermarked data, and negatives are generations from a model fine-tuned on the non-watermarked data. It achieves **46.5% accuracy**, i.e., close to random guessing. In other words, the blind baseline fails: it cannot tell whether a model was trained on the watermarked or non-watermarked dataset by looking only at n-gram statistics of its outputs, suggesting that any key-agnostic “micro-dialect” effect is very weak.
>
> **Q2: How does detection behave under overlapping keys?**
>
> **What we already show.** Table 6 already covers the basic two-owner case: a model trained on two datasets with two independent keys.
>
> **Same-domain stress test.** To match the reviewer’s “overlapping keyed corpora” scenario, we further split Alpaca into two halves D1, D2, watermarked them with $k_1$, $k_2$, and fine-tuned on the union. Per-key detection yields p-values: **6.6e-13 for $D_1$ on $k_1$** and **1.4e-33 for $D_2$ on $k_2$**. Both keys produce extremely small p-values on both halves, meaning cross-key detections are also strongly positive.
>
> **Q3: Should additional baselines be included?**
>
> Thanks for the suggestion.
>
> **STAMP-style.** STAMP-style methods are open-box and violate our strict black-box threat model (detector only has input-output access). A direct head-to-head is therefore not meaningful; we will clarify this mismatch.
>
> **Waterfall.** Comparison is meaningful only under quality constraints. TRACE targets distortion-free rewriting, whereas Waterfall significantly degrades text quality to achieve detection ability. Using the same hyperparameters as the original paper, Waterfall causes large perplexity (PPL) inflation under OPT-2.7B:
>
> | Perplexity | GSM8k | Med  | Dolly | Alpaca |
> |------------------------|-------|------|-------|--------|
> | SynthID                | 9.8   | 8.4  | 10.7  | 7.3    |
> | Waterfall              | 12.6  | 12.8 | 19.3  | 13.1   |
>
> Since TRACE operates in a low-distortion regime, Waterfall is not an appropriate baseline.
>
> **Q4: Will creators accept the rewrites?**
>
> We agree that acceptability to authors and creators is important.
>
> **Design philosophy.** TRACE is a protection mechanism, not a content-improvement system. While rewrites serve as training data, we ensure semantic integrity is preserved. Our goal is to maintain downstream task performance comparable to the original corpus—the key utility metric for model owners. Creator preferences vary; TRACE offers an option for those prioritizing protection and auditability.
>
> **Empirical quality check.** To directly address user preference, we conducted an LLM-as-a-judge evaluation (GPT-4o-mini). Using pairwise comparisons (AB/BA swap) on 200 Med samples, rewrites achieved a **0.77 win rate** on naturalness, fluency, and clarity. For style-sensitive domains (e.g., arXiv), we added a Creative-Fidelity judge and observed **0.98 agreement**, corroborated by embedding-based style similarity.
>
> [1] Blind baselines beat membership inference attacks for foundation models, IEEE SPW 2025
> [2] Scalable watermarking for identifying large language model outputs, Nature 2024

---

### Note · Authors · 2025-12-26

I have read and agree with the venue's withdrawal policy on behalf of myself and my co-authors.